# A scoping review of Youth Mental Health First Aid for adolescents in school, community, and healthcare settings

**Irfanul Alam**[1]*, **Marie Barnard**[2], **Jessica Osborne**[1],
**Divya Chandran Geetha Kumari**[3], **Clyde King Jr.**[3], **M. Allison Ford**[3],
**Hannah K. Allen**[3,4], **Sara J. Hendrix**[5], **Gracie Avery**[5], **Guadalupe Alvarado**[6]

**1** Center for Research Evaluation, The University of Mississippi, Mississippi, United States of America,
**2** Department of Public Health, The University of Mississippi, Mississippi, United States of America,
**3** Department of Pharmacy Administration, The University of Mississippi, Mississippi, United States of
America, **4** William Magee Institute for Student Wellbeing, The University of Mississippi, Mississippi,
United States of America, **5** Department of Health Science, The University of Alabama, Tuscaloosa,
Alabama, United States of America, **6** Substance Use and Mental Health Laboratory, The University of
Arkansas, Arkansas, United States of America

☺ These authors contributed equally to this work.
* iral3889@colorado.edu

pmen.0000549

Singapore, SINGAPORE

**Peer Review History:** PLOS recognizes the
benefits of transparency in the peer review
process; therefore, we enable the publication
of all of the content of peer review and
author responses alongside final, published
articles. The editorial history of this article is
available here: https://doi.org/10.1371/journal.
pmen.0000549

## Abstract

Youth Mental Health First Aid (YMHFA) is a training program that prepares adults
to recognize and respond to adolescents' mental health concerns. Although widely
used in schools and community organizations internationally, there has been no
recent synthesis focused on post-2019 implementation and effectiveness evidence in
youth-serving settings. This scoping review examined the extent, range, and charac-
teristics of the predominantly U.S.-based evidence on YMHFA in youth-serving set-
tings. We conducted a comprehensive scoping search of six bibliographic databases
and relevant gray literature sources through February 2025. Thirty-one studies met
the inclusion criteria. YMHFA was delivered in schools, libraries, youth nonprofits,
and family networks through in-person, blended in-person, and virtual formats. Most
studies reported short-term gains in adult participants' mental health literacy, confi-
dence, and stigma reduction. Barriers included scheduling conflicts, stigma-
related reluctance, staff turnover, and limited mental health infrastructure. Facilitators
included strong organizational leadership, funding, culturally tailored adaptations, and
community partnerships. Few studies assessed long-term outcomes, implementation
fidelity, or adolescent-level effects. Findings underscore the need for evaluations that
address sustainability, equity, and the direct impact of YMHFA on youth.

**Data availability statement:** All data relevant to this study are included in the manuscript and its supporting information files. No additional raw data were generated.

**Funding:** This work was supported by the Office of National Drug Control Policy (ONDCP; https://www.whitehouse.gov/ondcp/) under grant numbers CDS9923G0006 and CDS9924G0017 to HA.

**Competing interests:** The authors have declared that no competing interests exist.

## Introduction

Youth Mental Health First Aid (YMHFA) is an evidence-based training program that improves mental health literacy, enhances early intervention capabilities, and strengthens crisis response skills for individuals who interact with young people. The program equips adults with the knowledge and skills to recognize signs of adolescents' mental health challenges and to provide appropriate initial support until professional help is available.

YMHFA was developed in Australia in the mid-2000s as an adaptation of the Mental Health First Aid program, with a specific focus on supporting adolescents experiencing mental health challenges or crises. Since then, users have disseminated YMHFA internationally and widely adopted it in the United States across schools, community organizations, healthcare settings, and other youth-serving contexts. The YMHFA program structures itself around the ALGEE action plan, which provides a practical framework for responding to youth mental health concerns: assessing risk of harm or suicide, listening nonjudgmentally, giving reassurance and information, encouraging appropriate professional help, and promoting self-help and other support strategies.

YMHFA training is most often delivered as standardized, instructor-led courses for educators, parents, school staff, and community members [3]. Evaluations of YMHFA have frequently examined outcomes such as mental health literacy, confidence in recognizing and responding to youth mental health concerns, attitudes toward mental illness, stigma reduction, and self-reported helping behaviors [1–5]. Recent implementations have also explored adaptations to delivery formats, including virtual or hybrid training models, and cultural tailoring to better meet the needs of specific communities [6–8].

YMHFA has been implemented in a variety of settings across the United States, including schools, rural communities, and youth-serving organizations. Despite its increasing adoption, studies show that challenges such as mental health stigma, a limited number of mental health professionals in rural areas, and logistical barriers can hinder program delivery and affect reported outcomes [9,10]. Conversely, strong institutional support, school-based integration, and community partnerships serve as key facilitators that improve YMHFA's sustainability and impact [11,12]. While current research offers valuable insights, the variation in implementation contexts highlights the necessity of synthesizing evidence more effectively across different settings.

A preliminary search of MEDLINE, the Cochrane Database of Systematic Reviews, and JBI (Joanna Briggs Institute) Evidence Synthesis identified one prior systematic review focused on Youth Mental Health First Aid [3]. That review synthesized findings from studies published through 2019 and emphasized early evidence on knowledge and attitude changes among educators and school staff. However, it did not include more recent evaluations or broader implementation patterns across diverse populations and community-based settings. No scoping reviews were identified.

A scoping review methodology was chosen as the most suitable approach for this study due to the breadth and heterogeneity of the YMHFA literature and the exploratory nature of the research questions. Since the 2019 systematic review,

researchers have expanded their focus on YMHFA beyond early effectiveness studies to include diverse study designs, implementation evaluations, culturally adapted programs, and virtual or hybrid delivery models across various settings and populations. Notably, the COVID-19 pandemic has accelerated the shift toward remote and hybrid training formats [8,13], which has introduced new implementation contexts, barriers, and facilitators that make direct comparison using traditional systematic review methods more difficult.

Given the variability in context, methodology, and outcome measurement, a scoping review allowed us to map the extent, range, and characteristics of the available evidence without applying restrictive inclusion criteria or prioritizing a single outcome domain. Instead of synthesizing effect sizes or assessing study quality, this review aimed to characterize implementation strategies, identify reported barriers and facilitators, and summarize the outcome domains assessed across studies. Specifically, we asked: What literature currently exists on Youth Mental Health First Aid (YMHFA) for adolescents aged 12–18? Within that literature, what are the primary findings related to implementation and effectiveness?

In this review, YMHFA refers to the structured training program that equips adults with the knowledge and skills to support youth experiencing mental health challenges or crises [14]. Mental health literacy is defined as knowledge and beliefs about mental health conditions that support recognition, management, and prevention, consistent with how this construct is commonly operationalized in YMHFA evaluations [1,3]. Implementation is understood as the process by which YMHFA training is adopted, delivered, adapted, and sustained in community and institutional settings, including features such as delivery format, instructor model, organizational context, and participant engagement. Effectiveness is pragmatically defined as the extent to which YMHFA achieves its intended outcomes as reported in the literature, including changes in knowledge, confidence, attitudes, stigma, and helping behaviors among participants.

This review aimed to map how effectiveness has been conceptualized and measured across studies, alongside reported implementation processes, rather than to assess causal effectiveness or clinical outcomes. It also examined reported barriers and facilitators as factors that hindered or supported successful implementation and delivery. These constructs were used to map the scope of the evidence, characterize dominant outcome domains, and identify gaps and priorities for future YMHFA research and implementation.

## Methods

This scoping review was conducted according to the JBI methodology for scoping reviews and followed the PRISMA-ScR (Preferred Reporting Items for Systematic reviews and Meta-Analyses extension for Scoping Reviews) reporting guidelines [15,16]. The review protocol was deposited in an open-access repository (S1 appendix).

### Inclusion criteria

**Eligibility.**  Studies were eligible if they involved adolescents aged 12–18 or adults who directly supported or interacted with adolescents in this age range, including educators, school counselors, healthcare providers, coaches, caregivers, and youth workers. Studies in which adolescents served as peer supporters within YMHFA programs were also included, provided the primary focus was adolescent mental health. Studies were not excluded based on gender, ethnicity, race, socioeconomic status, or health condition. Excluded were studies focused exclusively on adults or on children younger than 12 without explicit relevance to adolescents.

**Concept.**  The concept under investigation was YMHFA, a structured educational intervention that teaches individuals to recognize signs and symptoms of adolescent mental health issues, provide initial support, and direct adolescents toward appropriate professional help or self-care strategies. Eligible studies addressed one or more elements of YMHFA, including implementation processes or strategies, evaluations of effectiveness, feasibility, or outcomes such as knowledge, attitudes, confidence, skills, behaviors, or related impacts on adolescent mental health. Studies that examined only general Mental Health First Aid (MHFA) without adaptation or application to adolescents were excluded.

**Context.** Studies were included if they were conducted in settings relevant to adolescents, such as schools, community organizations, after-school programs, clinics, hospitals, or virtual/online platforms. The geographic context was not restricted. Studies examining cultural, racial/ethnic, gender-specific, socioeconomic, or subcultural factors relevant to YMHFA delivery and outcomes were explicitly included. Excluded were purely theoretical, opinion-based, or commentary articles without empirical data, as well as studies exclusively focused on adult contexts.

## Source

A broad range of evidence sources was included. Eligible designs encompassed experimental and quasi-experimental studies such as randomized controlled trials, non-randomized trials, before-and-after studies, and interrupted time-series, as well as analytical observational studies like cohort, case-control, and analytical cross-sectional designs. Descriptive studies, including case series, descriptive cross-sectional studies, and case reports, were also considered, along with qualitative research, program evaluations, and mixed-methods studies. Knowledge syntheses, such as systematic reviews, scoping reviews, and rapid reviews, were eligible if they fulfilled the PCC criteria. To provide context on early YMHFA research and the evolution of strategies and outcomes, one prior systematic review was included [3]. Gray literature sources comprised policy documents, reports from government and NGO organizations, dissertations, theses, conference proceedings, and unpublished program evaluations from entities such as WHO, Mental Health First Aid International, SAMHSA, and NIMH.

**Search strategy.** A three-step search approach was employed. Initially, a preliminary search of MEDLINE (PubMed) was conducted to identify relevant articles, and keywords and index terms from these records were used to develop a comprehensive search strategy. The strategy was then adapted for each database (MEDLINE, PsycINFO, ERIC, CINAHL, Scopus) and gray literature source (S1 Checklist). Finally, the reference lists of included sources were screened for additional studies. Searches were completed in February 2025 and were limited to full-text publications in English. No date restrictions were applied.

**Source selection.** All citations were compiled in Zotero and deduplicated. Before formal screening, the review team conducted a pilot screening exercise to calibrate interpretation of the inclusion criteria and ensure consistency across reviewers, consistent with recommended scoping review practices [15,16]. Titles and abstracts were then screened independently by two or more reviewers against the eligibility criteria. Full texts of potentially relevant articles were retrieved and independently assessed by at least two reviewers, with reasons for exclusion documented. Discrepancies at any stage were resolved through discussion and consensus, with consultation from an additional reviewer as needed. The screening process is summarized in the PRISMA-ScR flow diagram (Fig 1 and S2 Appendix).

**Data extraction.** Data were extracted using a structured Qualtrics-based form (S3 Appendix) by two or more independent reviewers. Extracted fields included participant characteristics, study context, design, implementation processes, reported barriers and facilitators, and YMHFA-related outcomes. Following the extraction, the lead author conducted an iterative review of the charted data to ensure completeness and consistency across studies. Any discrepancies or uncertainties identified during extraction or synthesis were resolved through team discussion.

**Data charting and synthesis.** Extracted data were organized and tabulated according to concept, context, study design, implementation processes, reported barriers and facilitators, and program outcomes. A narrative synthesis accompanied the tabulated results. Four analytic domains were established a priori to guide synthesis: (1) Implementation Strategies, (2) Program Effectiveness, (3) Barriers to Implementation, and (4) Facilitators to Implementation.

During data charting, findings from each study were summarized within the appropriate domain to enable cross-study comparison. This approach identified shared patterns, context-specific variations, and notable implementation practices. Consistent with scoping review methodology and reporting guidance [15,16], formal inter-rater reliability statistics were not calculated because the review's objective was to map the scope and characteristics of the literature rather than to assess study quality or estimate pooled effects [17].

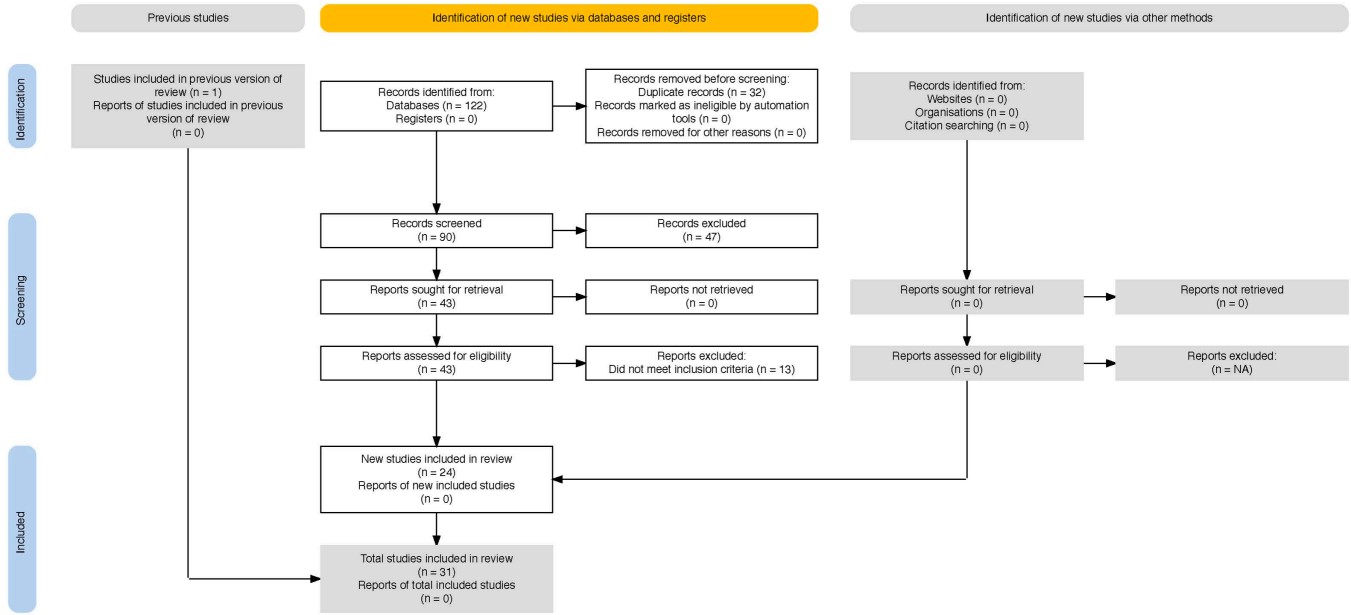

**Fig 1. The diagram illustrates the number of records identified, screened, excluded, and included at each stage of the scoping review process.**

## Study selection

A total of 122 records were identified through database searches. After removing 32 duplicates, 90 records remained for title and abstract screening. Of these, 43 full-text articles were assessed for eligibility. Thirteen articles were excluded for not meeting the inclusion criteria. Ultimately, 31 studies were included in the review, comprising 24 newly identified studies and 7 identified through prior mapping. Of the 31 included studies, 30 were primary research studies and one was a systematic review.

## Characteristics of included sources of evidence

The review included 31 sources of evidence published between 2011 and 2025. Most studies were conducted in the United States (n = 28), with additional studies from Australia (n = 4) and one multi-country systematic review covering the United States, Australia, and the United Kingdom (Table 1). The included studies varied substantially in design, setting, and delivery modality; implications of this heterogeneity for interpretation are examined in the Discussion. Study designs varied and included 17 quantitative studies (e.g., [1,10]), six mixed-methods studies [13,28], five program evaluations [11,19], two qualitative studies [31,33], and one systematic review [3].

Most studies focused on school-based implementations of YMHFA (n = 21; [4,24]). Others took place in community settings (n = 8; [29,36]) or involved partnerships between schools and community organizations (n = 2; e.g., [19,25]). Participant groups included educators, school staff, counselors, mental health professionals, social service employees, parents or caregivers, youth-serving professionals, and community members. Some studies also engaged non-traditional implementers such as law enforcement officers [35] and Cooperative Extension volunteers [32].

Sample sizes ranged from small pilot programs to large-scale statewide evaluations. Settings included public and private schools, after-school programs, community centers, healthcare facilities, and digital platforms. Many studies reflected real-world implementation contexts and described adaptations to meet participant or community needs. These adaptations included bilingual delivery [6,7], virtual formats [13,14], cultural tailoring [33,36], and integration into faith-based [36]

**Table 1. Summary of included studies reporting on YMHFA programs targeting adolescents across various settings (sorted by year).↑= increase;↓= decrease. NR=not reported in the source study; NA=not applicable to that study or outcome.**

| Citation | Country | Study Design | Source Type | Study Settings | Participants | Sample Size (N) | Gender distribution (%) | Outcomes | Facilitators | Barriers | Additional Context |
|---|---|---|---|---|---|---|---|---|---|---|---|
| [18] | Australia | Quantitative | Peer-reviewed journal article | Community (youth centers, clubs, extracurricular settings) | Community members | 246 | Female (76.4%), Male (23.6%) | ↑literacy, confidence, and intervention willingness; ↓stigma | High-quality or evidence-based training; Structured instructor training and certification process; National dissemination with public and private sector buy-in | Uncontrolled design limits causal inference; High attrition at 6-month follow-up; Limited generalizability due to self-selected participants with prior exposure to mental health issues | First published evaluation of YMHFA's 14-hour course. Demonstrates significant short- and medium-term gains in knowledge, confidence, and reduced stigma. Lays foundation for widespread dissemination and future controlled trials. |
| [1] | United States | Quantitative | Peer-reviewed journal article | Community | Social service employees | 384 | Female (73%) | ↑literacy, confidence, and intervention willingness | High-quality or evidence-based training | Time constraints during training sessions; Lack of control group; Post-test timing immediately after training | Pioneering U.S.-based study of YMHFA with strong pre-post evaluation using vignette-based assessment scored against the ALGEE framework. Demonstrates gains in applying ALGEE strategies and increased confidence, though limited by immediate post-test timing and absence of long-term follow-up. |
| [19] | United States | Program evaluation | Conference abstract | Schools (middle or high schools) and Community | Educators; Mental health providers; District administrators; Community members | 231 | NR | ↑literacy, behavior change, and intervention willingness | Supportive leadership or administration; Strong school-community partnerships; Integration into existing school or organizational systems | High student referral volume; Coordination challenges between school and community mental health; Sustainability concerns | Offers survey data from multiple stakeholders groups and highlights systemic challenges and early indicators of access improvements. Included hiring school-based mental health professionals, building triage and referral systems, and initiating mental health services. Descriptive report from a symposium presentation. |
| [4] | United States | Quantitative | Peer-reviewed journal article | Schools | Educators; Staff; Counselors | 356 | Female (83%), Male (17%) | ↑literacy, confidence, willingness to intervene, behavior, and attitudes ↓stigma | The data was provided by a larger study aiming to improve school climate and safety. Community social workers received instructor training and provided YMHFA instruction to staff at the chosen schools. | The researchers claim a lack of a control group in this study. Post test could have been given at a different time other than right after the training was provided. Participants noted that the 8-hour training was a limitation for them. There was also an inability to ensure fidelity to the YMHFA material. | Strong pre-post evidence of improvement, especially among staff with no prior training. Suggests YMHFA should be part of a broader behavioral health strategy in schools. |

*(Continued)*

**Table 1.** (Continued)

| Citation | Country | Study Design | Source Type | Study Settings | Participants | Sample Size (N) | Gender distribution (%) | Outcomes | Facilitators | Barriers | Additional Context |
|---|---|---|---|---|---|---|---|---|---|---|---|
| [10] | United States | Quantitative; Quasi-experimental | Peer-reviewed journal article | Schools | Educators; Staff, Para-professionals; Social workers; Counselors; Clinicians; Nurses | 205 | Female (80%), Male (19%), Missing (1%) | ↑literacy, confidence, and behavior | Not addressed | Not addressed | Highlights substantial gains in mental health literacy and confidence among non-mental health workforce aiders, particularly paraprofessionals. Suggests need for differentiated training for mental health professionals and supports paraprofessional delivery of brief interventions in schools. |
| [20] | United States | Mixed method | Peer-reviewed journal article | Schools | Educators; Athletic coaches; Staff | 149 | NR | ↑literacy, confidence, intervention willingness, and behavior; ↓stigma | High-quality or evidence-based training; Accessible school-based delivery; Direct student interaction opportunities | No baseline measures; Limited follow-up response rate; Lack of student-level data; Inconsistent timing between training and follow-up; Gaps in referral behavior despite training gains | Study shows positive reception and increased confidence/knowledge after YMHFA training among school staff. However, follow-up revealed reluctance or barriers in making referrals, and few conversations occurred around suicidal ideation. Highlights the gap between training and real-world application in school settings. |
| [2] | United States | Quantitative | Peer-reviewed journal article | Schools | Educators; Youth-serving professionals; School staff; Community members | 1244 | Female (82.2%), Male (17.3%) | ↑literacy, confidence, intervention willingness, and behavior; ↓stigma | High-quality or evidence-based training; Statewide coordinated implementation; Longitudinal follow-up at 3, 6, 9, and 12 months; Analysis of demographic subgroups | High attrition at later follow-up points; Self-selection bias; No control group; Limited fidelity monitoring | Found sustained improvements in knowledge, confidence, and ALGEE use. Suggests broad reach across demographic groups but emphasizes need for stronger fidelity of data and participant retention. Strong emphasis on YMHFA integration into school Multi-Tiered System of Support models |
| [21] | United States | Quantitative; Quasi-experimental | Peer-reviewed journal article | Schools | Graduate social work students | 73 | Female (85%), Male (15%) | ↑attitudes toward youth mental health, mental health literacy and self-confidence in applying ALGEE | Integration into field education; Instructor-led, no-cost training; Practical utility for client work | Self-selection bias; Limited generalizability; Stigma scale reliability issues | Among the first U.S. quasi-experimental trials of YMHFA USA with graduate social work students, interpreted through the lens of the Unified Theory of Behavior, demonstrating enduring positive changes over time |

*(Continued)*

**Table 1.** (Continued)

| Citation | Country | Study Design | Source Type | Study Settings | Participants | Sample Size (N) | Gender distribution (%) | Outcomes | Facilitators | Barriers | Additional Context |
|---|---|---|---|---|---|---|---|---|---|---|---|
| [5] | United States | Quantitative | Peer-reviewed journal article | Schools and Community | Child/Youth-serving professionals | 893 | Female (83%), Male (17%) | ↑ confidence to intervene/preparedness, mental health literacy, satisfaction with the program | The data from this study would further contribute to YMHFA research. Diverse sample sector-specific analysis. 12 countries would go on to host their own YMHFA training. | No long-term follow-up; self-report data only; Outcomes were assessed only immediately post-training | Large sample spanning four occupational sectors enhances generalizability of findings. Strong evidence for effectiveness and high satisfaction across sectors supports YMHFA's universality. |
| [22] | Australia | Quantitative; Randomized controlled trial (RCT) | Peer-reviewed journal article | Schools and Community | Parents or caregivers | 322 | Female (88.2%) | ↑ literacy, confidence, intervention willingness, and behavior | High-quality or evidence-based training; Use of an active control group; Longitudinal assessment | Attrition over time; Underpowered to detect changes in aid recipients; Limited evidence of behavioral change despite knowledge gains | Longest follow-up of a controlled trial of YMHFA to date. Significant knowledge gains were sustained over 3 years. However, the study lacked power to detect strong behavioral or recipient outcomes, underscoring the need for refresher trainings and improved behavioral measurement tools. |
| [23] | United States | Quantitative; Quasi-experimental | Peer-reviewed journal article | Schools and Community | Youth-serving professionals | Pre-post=987 Follow-up = 238 | Pre-post=Female (81.9%), Male (18.1%) Follow-up = Female (91.2%), Male (8.8%) | ↑ literacy, confidence, intervention willingness; sustained at 90 days | Analysis of individual-level characteristics; Three timepoint design | Ceiling effects for high pre-training scorers; No behavioral outcome measures; Lack of contextual job role data | Showed YMHFA is more effective for participants with lower pre-existing attitudes and knowledge. Suggests cost-effective implementation strategies through tiered or targeted approach to YMHFA implementation based on workforce characteristics. |
| [3] | United States, Australia, United Kingdom | Systematic review | Peer-reviewed journal article | Schools | Educators; school staff; community members; youth-serving professionals | NA | NA | ↑ literacy, confidence, and intervention willingness; ↓ stigma; limited long-term behavior change | High-quality or evidence-based training; inclusion of ALGEE model; school-based implementation | Limited long-term follow-up; variation in training delivery and fidelity; heterogeneity in outcome measures | Synthesizes findings from 8 YMHFA studies through 2019. Identifies consistent improvements in knowledge and attitudes across settings but limited behavioral data and generalizability. Calls for more RCTs and culturally responsive adaptations. |
| [24] | United States | Mixed method | Dissertation | Schools | Educators; School counselors | 176 | Female (52.3%), Male (9.1%), Not answered (38.6%) | ↑ literacy, confidence, intervention and willingness | This intervention was conducted in McHenry Count schools in Illinois State-level training mandate; County-level funding; Trainer model using school staff; Relevance to crisis context | Time constraints; Competing priorities; Staff turnover; Stigma; Lack of follow-up or fidelity tracking. COVID-19 pandemic limited the training opportunities. | Highlights training's perceived effectiveness while noting the need for sustainability and long-term outcome data. |

*(Continued)*

**Table 1.** (Continued)

| Citation | Country | Study Design | Source Type | Study Settings | Participants | Sample Size (N) | Gender distribution (%) | Outcomes | Facilitators | Barriers | Additional Context |
|---|---|---|---|---|---|---|---|---|---|---|---|
| [25] | United States | Mixed method | Peer-reviewed journal article | Schools | Parents or caregivers | Parents=107 (pre/post), 64 (2-month follow-up) | NR | ↑literacy, confidence, help-seeking intentions, attitudes, behavioral intentions; ↓stigma | Interactive delivery and accessibility through school-based offerings (supported by thematic feedback) | Not addressed | One of the first studies to engage parents in YMHFA through schools, underscoring the importance of developing parent-tailored modules and leveraging parent-school collaboration |
| [6] | Australia | Mixed method | Peer-reviewed journal article | Schools | Adolescents; Educators; Caregivers or parents | Adolescents=308 Adults=34 | Adolescents=Female (50.5%), Male (48.2%) Adults=Female (72%), Male (28%) | ↑literacy, confidence, knowledge, intervention willingness, and attitudes; ↓ stigma | Cultural tailoring; bilingual delivery; community engagement | Minor logistical issues with session scheduling and follow-up survey completion. This study used an uncontrolled design. | First culturally adapted teen and YMHFA training tailored for ethnically diverse adolescents and adults in Australia. Notable for use of culturally tailored scenarios, Culturally and Linguistically Diverse (CALD) instructors, and collaboration with local services. Youth were more willing to seek help from adults after post-training, especially important for CALD youth who were hesitant to disclose mental health issues. |
| [7] | United States | Mixed method | Peer-reviewed journal article | Community | Parents and youth-serving professionals | Parents =31 Youth workers=24 | Parents=Female (71%), Male (29%) Youth workers=Female (87.5%), Male (12.5%) | ↑mental health literacy, confidence, and helping intentions | Cultural relevance; Community engagement; Bilingual delivery; Inclusion of culturally specific risk/protective factors | Cultural stigma; Limited generalizability (highly educated, mostly Chinese participants) | First culturally adapted YMHFA evaluated; added pre-session and Asian-culturally tuned curriculum; follow-up focus groups emphasized the need for scenario modeling and ongoing booster sessions. |
| [26] | United States | Quantitative; Program evaluation | Peer-reviewed journal article | Schools | Teachers; counselors; psychologists; social workers; nurses; Paraprofessionals; Administrators | 73 | NR | ↑confidence, preparedness, likelihood to refer | Supportive leadership or administration; Strong school-family or community partnerships; Integration into existing school systems | Time constraints or scheduling difficulties; Physical space limitations; Mental health stigma; Belief that schools are not ideal for mental health services; Competing academic priorities | Collaborative evaluation with extensive stakeholder involvement; Staff perceived improvements in school climate and preparedness. |
| [8] | United States | Quantitative | Peer-reviewed journal article | Community | Community members | 35 | Female (85.7%) | ↑mental health literacy, knowledge, confidence, help-seeking attitudes and intentions; ↓stigma; | Culturally adapted content; Community engagement; Interactive format | No behavioral or long-term follow-up; Sample skewed toward Chinese Americans with high education; No control group | First U.S. study of culturally adapted virtual YMHFA for Asian Americans. Pre–post survey showed large gains in literacy, knowledge, and confidence. Supports culturally tailored, accessible parent training formats. |

*(Continued)*

**Table 1.** (Continued)

| Citation | Country | Study Design | Source Type | Study Settings | Participants | Sample Size (N) | Gender distribution (%) | Outcomes | Facilitators | Barriers | Additional Context |
|---|---|---|---|---|---|---|---|---|---|---|---|
| [27] | United States | Quantitative | Peer-reviewed journal article | Schools | University students, faculty and staff; Public school teachers and staff | 293 | Female (75.93%); Male (24.07%) | ↑ self-reported use of ALGEE helping behaviors across 3–6 months; behavior declined by 9 months | Repeated follow-up measurements | Course attrition over time; Self-report bias; Curriculum variation between youth and adult MHFA; Limited demographic diversity | Supports the need for booster trainings and tailored curriculum adjustments. |
| [28] | United States | Mixed Methods | Peer-reviewed journal article | Schools | Educators | 106 | Female (79%), Male (15%), Not reported (5%) | ↑ mental health literacy, confidence, knowledge, and behavior; no change in stigma; mixed retention at 3-month follow-up | Cultural adaptations; Relevant and digestible content; Use of real-life scenarios; High satisfaction | Stigma; Lack of linguistic/cultural specificity; One-size-fits-all perception of ALGEE; Need for boosters; Limited coverage of schizophrenia; Follow-up attrition | First U.S. study of YMHFA with AmeriCorps educators. Found significant gains in MHFA behaviors and knowledge with cultural adaptations, though stigma remained unchanged. Highlights need for booster sessions and tailored supports for CALD students. |
| [29] | United States | Mixed method | Peer-reviewed journal article | Community | Community members | 2314 | Female (80.8%) | ↑ literacy, confidence, intervention willingness; ↓ stigma | High-quality or evidence-based training; Large-scale internal evaluation; Theory-based measures; Real-world implementation data | Course length or pacing; Material inconsistencies across manuals and slides; Underrepresentation of key demographic groups; Lack of long-term behavioral outcome data | Largest internal evaluation of YMHFA and Adult MHFA in the U.S. to date. Strong evidence of knowledge and confidence gains. Suggests refinements to course materials, pacing, and expanded outreach to underrepresented groups. |
| [11] | United States | Program evaluation | Peer-reviewed journal article | Schools | Educators | 99 | Female (83%), Male (6%), Did not respond (11%) | ↑ knowledge, attitudes, and confidence | SAMHSA funding, university-school partnerships, tailored implementation, free access | Rural access issues, sustainability concerns, limited generalizability, no control group | Describes a SAMHSA-funded YMHFA initiative in West Virginia rural schools. Highlights structured planning, strong outcomes, and context-specific delivery strategies. Notes importance of culturally relevant adaptations, local partnerships, and follow-up supports. |
| [14] | United States | Quantitative | Peer-reviewed journal article | Schools and Community, Online and Digital | Educators; School staff; Community members; Mental health professionals | 480 | NR | ↑ confidence in recognizing and supporting youth mental health needs | In-person training delivery; Targeting participants with less mental health background | Limited long-term follow-up; Differences in training delivery not randomized; Instructor variability not assessed | Robust California-based study comparing trained vs. untrained participants. Demonstrates greater gains for in-person training and those with less mental health experience. Highlights confidence increases across all groups and supports strategic targeting of training. |

*(Continued)*

**Table 1.** (Continued)

| Citation | Country | Study Design | Source Type | Study Settings | Participants | Sample Size (N) | Gender distribution (%) | Outcomes | Facilitators | Barriers | Additional Context |
|---|---|---|---|---|---|---|---|---|---|---|---|
| [30] | United States | Quantitative; Quasi-experimental | Peer-reviewed journal article | Schools | Teachers; School staff; Administrators | 843 | Female (79.3%), Male (20%) | ↑knowledge, confidence, attitudes, and willingness to intervene | Supportive leadership or administration; High-quality or evidence-based training; Instructor involvement or certification support; Use of flexible or blended training formats; Cultural relevance or adaptations | Limited availability of mental health professionals; Cultural or language barriers; Inadequate post-training support or follow-up | Focused on culturally responsive YMHFA implementation in Catholic schools serving Latinx youth in South Texas. Offers rich quantitative and contextual insight for implementation in religious-affiliated educational settings. |
| [31] | United States | Qualitative | Peer-reviewed journal article | Community | Extension educators | Survey=40 Interview=11 | NR | Perceived increases in staff confidence, knowledge-sharing, program reach; raised awareness of structural implementation needs | Strong partnerships, secure funding, dedicated coordination, organizational support | Staffing limitations, funding insecurity, technology/connectivity constraints, rural access challenges | Rich qualitative analysis of YMHFA delivery across 11 states via Cooperative Extension. Highlights structural, cultural, and systemic implementation supports and barriers. Suggests clear strategies for sustaining and scaling up YMHFA within community-based systems. No direct pre/post outcome measures reported. |
| [32] | United States | Mixed method; Program evaluation | Peer-reviewed journal article | Community | Extension volunteers; Youth-serving professionals | 99 | Female (83%), Male (6%), Did not respond (11%) | ↑confidence, beliefs, and supportive behaviors; greater increase for YMHFA in one outcome (talking with youth) | A study approved by the University of California. External partners recruited instructors to provide multiple trainings. Use of multiple training formats; Scaled reach via Extension; Participant flexibility; Cultural tailoring in alternative program design | Time commitment burden for YMHFA; Accessibility issues (e.g., broadband, scheduling); Limited diversity in pilot sample. Technology issues, cost, and accessibility. | Compares YMHFA and SYMH for adult volunteers in Extension. YMHFA showed stronger effect for one outcome (talking with youth), but SYMH was more accessible and acceptable. Supports SYMH as a viable gatekeeper training when YMHFA is infeasible. |
| [33] | United States | Qualitative | Peer-reviewed journal article | Community | Caregivers or parents; Healthcare providers; community members | Professionals:=10 Parents =6 | Professionals:=Female (90%) Parents=Female (83%) | ↑mental health literacy and confidence; ↓stigma | Cultural relevance or adaptations; Strong school-family or community partnerships | Cultural or language barriers; Persistent mental health stigma; Inadequate post-training support or follow-up, not enough participants recruited, | First study focused on culturally adapting YMHFA for South Asian American and Southeast Asian American parents. Emphasizes importance of cultural tailoring, shared community language, and integration of family-based values. Suggests practical recommendations like pre-training education, culturally matched trainers, and real-life scenario modeling for ALGEE. |

*(Continued)*

Table 1. (Continued)

| Citation | Country | Study Design | Source Type | Study Settings | Participants | Sample Size (N) | Gender distribution (%) | Outcomes | Facilitators | Barriers | Additional Context |
|---|---|---|---|---|---|---|---|---|---|---|---|
| [34] | United States | Qualitative; Program evaluation | Dissertation | Schools | Educators; School counselors; Youth workers | 18 | Female (100%) | ↑literacy, confidence, preparedness, willingness to intervene; behavior change | Supportive leadership; Funding/resources; High-quality training (ALGEE); Instructor certification support; Integration into school systems | Funding/resource gaps; Time constraints; Stigma; Lack of post-training support; Fidelity monitoring; Mandate ambiguity | Explores YMHFA post-state mandate in Florida public schools. Highlights inconsistent implementation and calls for standardized guidelines. |
| [13] | United States | Mixed method | Peer-reviewed journal article | Schools | Educators; school staff; student teaching interns; administrators | 36 | Female (80.6%), Male (19.4%) | ↑literacy, confidence, intentions to help and attitudes | Cultural alignment; Flexible, virtual access; Real-life scenarios; Interactive format; Acceptability across diverse roles | Scheduling conflicts; Limited content depth; Need for more cultural tailoring and booster sessions | Focused on immigrant-origin youth serving schools. Findings highlight promise of remote delivery with suggestions for content enhancements |
| [35] | United States | Quantitative | Peer-reviewed journal article | Community | Law enforcement officers (LEO) | 51 | Female (52.9%), Male (47.1%) | ↑literacy, confidence, intervention willingness, and attitudes; ↓stigma | The data from this study came from a larger project funded by the Substance Abuse and Mental Health Services Administration. This Administration provided YMHFA training to LEOs. | Low follow-up rate; Ceiling effects in baseline scores; No mention of time constraints, scheduling, or access issues. | First known longitudinal evaluation of YMHFA with law enforcement officers; demonstrates strong short-term gains and partial retention at 90 days, with insights into gender and role-based differences. Although there was improvement in attitudes and confidence, LEOs already scored high in knowledge of mental health, so they did not have much room for improvement. Overall, they were highly satisfied with the trainings. |
| [36] | United States | Mixed method | Peer-reviewed journal article | Community | Parents; Community members | Survey: 24 Interviews: 12 | Survey=Female (83%) Interviews=Female (92%) | ↑literacy, confidence, intervention willingness, behavior, self efficacy, and attitudes; ↓stigma | Approved by the Institutional Review Bord of the University – University of Alabama. Culturally grounded adaptation process; Korean-language materials; Korean-speaking facilitators; Faith-based partnerships; Small-group delivery model | Small sample size; Limited generalizability; Short follow-up period; Need for validated Korean measures | Pilot study evaluating a culturally adapted YMHFA for Korean American communities in the Southern U.S. Found significant improvements in confidence, knowledge, and stigma reduction. Community-based delivery and linguistic/cultural adaptations were critical. |

or immigrant-serving contexts [13]. Several recent studies emphasized culturally grounded approaches for Latinx, Asian American, and immigrant-origin communities [28,30,33]. Two studies directly involved adolescents as participants or peer supporters in YMHFA programming [6,34].

## Results

Our scoping review results are organized across four analytic domains: implementation strategies, program effectiveness, facilitators to implementation, and barriers to implementation. Together, these findings describe the scope, heterogeneity, and key patterns in the YMHFA literature across youth-serving settings.

### Implementation strategies

YMHFA was implemented in educational, community, and clinical settings using different formats, instructor approaches, and adaptation strategies. Most programs followed the standardized, evidence-based curriculum centered on the ALGEE action plan, although some made contextual modifications to address specific population needs, logistical challenges, or cultural considerations.

**Training formats and delivery modalities.** YMHFA most often delivers as an in-person, one-day (8-hour) course or across multiple sessions. For example, Marsico et al. [25] offer flexible schedules ranging from a single 8-hour session to four 2-hour segments, while Morgan et al. COVID-19 pandemic [8,13]. Virtual sessions often combine asynchronous prework with live training, improving access for working professionals and geographically dispersed participants.

**Instructor models and fidelity to curriculum.** Certified YMHFA instructors, often based in schools, public health departments, or community organizations, conduct all implementations. Partnerships with state agencies [14] and national organizations such as the Mental Health Association [1,21] support instructor consistency and fidelity to the curriculum. Most studies strictly adhere to the core curriculum, though some introduce cultural adaptations [7,8,28], such as community-specific risk factors, language accommodations, or culturally relevant examples.

**Participant recruitment and target audiences.** Educators [20,26], social workers [21], parents [7,25], and AmeriCorps members [28] participate. Recruitment strategies include school district outreach [34], professional networks [23], community partnerships [31], and public advertising [18]. Some programs rely on voluntary participation, while others incorporate YMHFA into mandatory professional development [20,30].

**Innovations and adaptations.** Role-play scenarios [28], parent-child dyad participation [22], and participatory cultural adaptation models [7] represent innovative practices. Programs serving Asian American parents or immigrant-origin youth educators emphasize trust-building, culturally resonant content, and community-informed facilitation [8,13].

**Institutional support and integration.** Implementation often depends on aligning with existing systems, such as professional development schedules [34], state or district leadership [14], or collaboration with school mental health staff [26]. Larger initiatives like Project AWARE [10,19] benefit from structured rollout plans, funding, and cross-sector collaboration.

### Program effectiveness

Across the reviewed studies, YMHFA consistently demonstrated positive effects on participants' mental health literacy, confidence in supporting youth, and attitudes toward mental health. Researchers primarily assessed effectiveness through pre- and post-intervention surveys, with some studies including qualitative feedback or follow-up assessments.

**Mental health literacy and knowledge gains.** Most studies reported significant improvements in participants' knowledge of youth mental health issues, including recognition of symptoms, risk factors, and intervention strategies. For example, Morgan et al. [22] found that parents gained a better ability to recognize mental health disorders, while Bhaktha et al. [29] and Noltemeyer et al. [2] reported increased knowledge among school staff and community participants.

**Confidence and intent to help.** Participants such as educators [26,34], parents [7], and youth-serving professionals [28] expressed greater confidence in discussing mental health and applying the ALGEE framework. In some cases [27], they acted on this confidence, leading to higher rates of helping behaviors in follow-up surveys.

**Reduced stigma and shifts in attitudes.** Several studies observed that participants reduced their stigma toward youth with mental health challenges. They became less fearful or judgmental toward individuals experiencing depression, anxiety, or suicidal thoughts [1,30]. Culturally adapted versions, such as Wang et al. [7], effectively addressed stigma among Asian American parents.

**Behavioral intent and application of skills.** Researchers measured participants' likelihood to intervene or refer youth after training. Ross et al. [28] found that AmeriCorps members became more likely to initiate conversations about mental health, while Kelly et al. [18] and Geierstanger et al. [14] reported increased referrals to counseling services and school-based providers.

**Sustained outcomes and follow-up assessments.** Few studies examined long-term outcomes. Morgan et al. [22] and Noltemeyer et al. [2] showed that gains in knowledge and confidence generally persisted at 3- and 12-month follow-ups, although intentions to help decreased without reinforcement. Laurene et al. [27] recommended booster sessions or follow-ups to maintain behavior change.

**Population-specific outcomes.** Effectiveness sometimes varied based on participant background. Laurene et al. [27] observed that school personnel made greater gains than community members, while Allert [20] reported larger knowledge increases among teachers. Culturally adapted trainings showed promise in addressing baseline gaps and cultural barriers [7,22].

## Facilitators of implementation

Organizational, cultural, and participant-level factors supported the implementation of YMHFA, shaping program reach, acceptability, and sustainability across settings.

**Organizational buy-in.** Programs achieved stronger implementation when organizations integrated YMHFA into existing school systems, state initiatives, or district-wide programming. Administrative leaders increased program legitimacy, allocated time and resources, and coordinated staff roles, which supported broader participation and sustained delivery [5,10,14].

**Flexible delivery formats.** Flexible delivery formats improved accessibility for participants managing professional and caregiving responsibilities. Programs that offered hybrid, asynchronous, or modular training formats accommodated educators, parents, and community members more effectively, particularly in rural or resource-constrained settings [13,25].

**Cultural tailoring.** Programs strengthened engagement when they aligned training content with participants' linguistic, cultural, and community contexts. Culturally familiar scenarios, bilingual delivery, and attention to culturally specific beliefs about mental health reduced stigma and increased participant comfort and perceived relevance [7,8].

**Instructor credibility.** Trainers played a critical role in implementation when they shared cultural, linguistic, or professional backgrounds with participants. This alignment fostered trust and relatability and encouraged open discussion of sensitive topics such as suicide and mental illness [20,28].

**Peer support and interactivity.** Programs reinforced skill development and participant confidence through peer support and interactive components such as role-play, group discussion, and collaborative learning. These strategies helped participants translate knowledge into practice and retain key concepts [28,31].

**Strategic partnerships.** Schools, community organizations, and health providers expanded dissemination and strengthened referral pathways by forming strategic partnerships. In settings with limited mental health infrastructure, cross-sector coordination supported program feasibility and reach [21,30].

**Participant motivation.** Participant characteristics shaped implementation outcomes. Individuals with prior exposure to youth mental health concerns or professional responsibilities related to youth support engaged more fully with training and completed follow-up activities at higher rates, indicating differences in readiness across participant groups [27,29].

   

## Barriers to implementation

Despite overall positive outcomes, studies identified persistent barriers related to logistics, organizational readiness, stigma, and evaluation capacity.

**Scheduling and logistical constraints.** Programs encountered significant scheduling challenges when delivering the standard 8-hour YMHFA training. Schools and families often struggled to accommodate the full training within existing schedules, prompting some programs to condense or split sessions, sometimes at the expense of participant engagement or fidelity to the curriculum [25,34]. Virtual delivery reduced some access barriers but introduced new challenges related to time-zone coordination, caregiving responsibilities, and participant availability [13].

**Recruitment and retention challenges.** Programs experienced difficulty recruiting and retaining parents and community members. Studies reported low turnout in some culturally specific communities due to logistical conflicts, mistrust, or competing obligations, while school staff frequently faced overlapping professional development requirements that limited sustained participation [7,8,20,21].

**Stigma and cultural beliefs.** Cultural stigma constrained open discussion of mental health and suicide, particularly in communities where these topics remain highly stigmatized. Such beliefs reduced willingness to participate in training and limited depth of engagement during sessions [7,13].

**Technology and infrastructure limitations.** Virtual and hybrid implementations faced barriers related to unreliable internet access, limited device availability, and low digital literacy. These constraints disproportionately affected rural and underserved settings and reduced the effectiveness of remote training formats [13,14].

**Trainer availability.** Limited availability of certified instructors restricted program scalability. Several studies identified instructor shortages and fatigue as factors that reduced the frequency and geographic reach of YMHFA trainings [28,30].

**Organizational misalignment.** Programs faced implementation challenges when YMHFA goals conflicted with institutional priorities or lacked policy support. In school settings, competing academic demands and unclear mandates delayed adoption, while community-based programs encountered unstable funding or misalignment with organizational missions [21,26,34].

**Evaluation challenges.** Evaluation limitations constrained assessment of longer-term outcomes. High attrition in post-training surveys and difficulty maintaining participant contact reduced the availability of follow-up data, limiting conclusions about sustained impact [2,22].

## Discussion

This scoping review examined how Youth Mental Health First Aid is implemented and how effective it is in youth-serving settings around the world. By synthesizing findings from various studies, four domains emerged: implementation strategies, program effectiveness, barriers, and facilitators. These domains highlight both the potential and the challenges of delivering YMHFA in real-world contexts.

In different settings, instructors ranging from school staff to mental health professionals and community leaders delivered YMHFA in multiple formats. They often adapted implementation strategies to meet local needs, especially regarding scheduling, delivery methods, and target audiences. Despite these variations, trainings consistently resulted in short-term benefits, such as increased mental health literacy, greater confidence in providing support, and reduced stigma. These outcomes appeared across educators, parents, school staff, and other youth-serving professionals.

Studies also identified ongoing implementation challenges: logistical issues like scheduling and retention, stigma-related barriers, limitations of virtual delivery, and gaps in institutional or policy alignment. On the other hand, successful implementation depended on organizational buy-in, flexible delivery formats, culturally tailored adaptations, and strong partnerships with community organizations.

This heterogeneity underscores the importance of interpreting YMHFA findings based on the study's purpose and context. While improvements in mental health literacy, confidence, and stigma were common, the extent, durability, and practical application of these outcomes varied depending on the setting, population, and implementation strategy. This pattern does

not indicate inconsistency in program effectiveness but shows how YMHFA is adapted and integrated into diverse real-world systems. Therefore, the main contribution of this scoping review is not to judge effectiveness narrowly but to clarify how and under what conditions YMHFA proves most feasible, acceptable, and sustainable in youth-serving environments.

### Interpretation and implications

These findings align with broader patterns in implementation science: the success of a program depends not only on the intervention but also on the systems and contexts that support it [3]. YMHFA achieves the best results when programs balance flexibility with fidelity. Tailoring delivery for parents, immigrant communities, or nontraditional learners increases engagement, but adaptations must be monitored to ensure that core components remain intact.

Organizational leadership and cross-sector collaboration play a consistently critical role. Programs embedded in larger initiatives, such as Project AWARE or district-level capacity-building, attain greater reach and sustainability. In contrast, ad-hoc trainings or efforts led by a single champion face a higher risk of discontinuation. These findings suggest that YMHFA thrives when it integrates into broader infrastructures with administrative and policy support.

Cultural and linguistic tailoring also proves vital. Studies serving Asian American parents and racially minoritized communities demonstrate that trust, shared identity between instructors and participants, and culturally familiar examples enhance participation and completion. Such results underscore the importance of participatory approaches that involve communities as partners rather than passive recipients.

### Interpreting findings across heterogeneous study designs and contexts

The YMHFA literature reviewed here exhibits substantial heterogeneity in study design, implementation context, and delivery modality, which has important implications for interpreting findings. Studies range from randomized controlled trials to quasi-experimental designs, program evaluations, and qualitative analyses. Controlled trials tend to provide stronger evidence for short-term changes in knowledge and attitudes, while program evaluations and mixed-methods studies offer critical insights into real-world implementation, feasibility, and sustainability. Although these types of evidence serve complementary purposes, they should not be interpreted as equivalent for causal inference.

Implementation context also influences reported outcomes and challenges. School-based implementations often benefit from existing professional development structures and access to youth but face constraints related to scheduling, staff turnover, and competing academic priorities. In contrast, community-based implementations frequently demonstrate flexibility and cultural responsiveness, especially in parent- and community-led models, but encounter challenges with recruitment, funding stability, and infrastructure. These contextual differences shape not only reported outcomes but also the barriers and facilitators observed.

Delivery modality adds to the heterogeneity in the evidence base. In-person training typically leads to higher engagement and more interactive learning opportunities, whereas virtual and hybrid formats expanded significantly during the COVID-19 pandemic, improving accessibility and reach. However, these formats also introduce challenges related to technology access, participant engagement, and fidelity to standardized content. Differences across delivery modes complicate direct outcome comparisons but reveal important trade-offs for implementation decision-making.

Overall, this heterogeneity highlights the importance of interpreting YMHFA findings in light of study purpose and context rather than viewing them as uniform indicators of program effectiveness. The diversity of evidence demonstrates YMHFA's adaptability across settings and emphasizes the need for future research to systematically examine how design choices, context, and delivery format influence outcomes.

### Conclusion

This scoping review synthesized evidence on the implementation and effectiveness of Youth Mental Health First Aid in youth-serving settings across the United States. The findings show that YMHFA is highly adaptable and generally well received, with short-term benefits such as improved mental health literacy, increased confidence, and reduced stigma.

Implementation is influenced by contextual barriers, including logistical, institutional, and cultural challenges, which require careful planning and adaptation.

Future implementation efforts should prioritize institutional commitment, culturally responsive approaches, and cross-sector partnerships to strengthen sustainability and reach. Research should go beyond short-term outcomes to examine long-term impacts on both participants and youth, systematically assess fidelity, and address gaps in underrepresented populations and settings. Advancing this work is essential to realize YMHFA's potential as a scalable, equity-oriented intervention in youth mental health.

### Gaps in the literature

Despite promising evidence, notable gaps remain. Most studies measure only immediate outcomes, with little evidence on long-term behavioral change or youth mental health impact. Few studies systematically assess fidelity of implementation or document adaptation processes. Populations such as Indigenous youth, LGBTQ+ adolescents, and rural communities remain underrepresented. Finally, little is known about the cost, scalability, or sustainability of YMHFA, which are critical considerations for widespread adoption.

A notable feature of the YMHFA evidence base is that most reported outcomes concern adults who interact with adolescents rather than adolescents themselves. This focus reflects YMHFA's design as a gatekeeper training aimed at improving adults' ability to recognize, respond to, and support youth experiencing mental health challenges. Accordingly, most assessed outcomes include adult mental health literacy, confidence, attitudes, stigma, and helping behaviors. Only a limited number of studies directly examine adolescent-level outcomes, such as youths' willingness to seek help, perceptions of adult support, or engagement with mental health services. Evidence on direct impacts on adolescents' mental health status remains sparse, partly due to ethical, logistical, and methodological challenges associated with measuring youth mental health outcomes in implementation-focused studies. This gap highlights the need for future research to systematically examine downstream effects of YMHFA on adolescents, including both intended and unintended outcomes.

### Limitations

This review has several limitations. We included only full-text studies in English, which may have excluded relevant evidence from other contexts or unpublished sources. Following scoping review methodology, we did not assess the quality of studies or the strength of the evidence. Additionally, the considerable variation in study designs, populations, settings, and outcome measures limited direct comparisons across studies.

Most included studies examined outcomes among adults supporting adolescents, with relatively limited evidence on outcomes at the adolescent level. Therefore, conclusions about YMHFA's downstream impact on youth mental health should be interpreted with caution.

### Future directions

Future research should prioritize longitudinal studies to determine whether YMHFA results in sustained behavior change and improved outcomes for youth, including direct adolescent-level mental health and help-seeking results. Researchers need to pay more systematic attention to fidelity and adaptation processes by using frameworks that document what is modified and why. Mixed-methods implementation studies that incorporate participant perspectives, policy environments, and delivery infrastructure would strengthen the evidence base. Finally, extending research into underrepresented communities, including rural, tribal, and housing-insecure populations, is essential to ensure equitable access to youth mental health supports.

## Supporting information

**S1 Appendix. Scoping review protocol.**
(PDF)

**S2 Appendix. Search strategy.**
(PDF)

**S3 Appendix. Qualtrics extraction form.**
(PDF)

**S1 Checklist. PRISMA-ScR checklist.**
(DOCX)

## Acknowledgments

We thank the Southern Mental Health Alliance for their support of Youth Mental Health.

## Author contributions

**Conceptualization:** Marie Barnard.

**Data curation:** Irfanul Alam.

**Formal analysis:** Irfanul Alam, Jessica Osborne, Divya Chandran Geetha Kumari, Clyde King Jr., Hannah K. Allen, Sara J. Hendrix, Gracie Avery, Guadalupe Alvarado.

**Funding acquisition:** Hannah K. Allen.

**Investigation:** Irfanul Alam.

**Methodology:** Irfanul Alam.

**Project administration:** Irfanul Alam.

**Supervision:** Irfanul Alam, Marie Barnard.

**Validation:** Irfanul Alam, Jessica Osborne.

**Writing – original draft:** Irfanul Alam.

**Writing – review & editing:** Irfanul Alam, Marie Barnard, Jessica Osborne, Divya Chandran Geetha Kumari, M. Allison Ford, Hannah K. Allen, Sara J. Hendrix, Gracie Avery, Guadalupe Alvarado.

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
