## [Decision Letter · Decision Letter 0]

3 Dec 2025

PMEN-D-25-00530

A scoping review of youth mental health first aid for adolescents in school, community, and healthcare settings.

PLOS Mental Health

Dear Dr. Alam,

Thank you for submitting your manuscript to PLOS Mental Health. After careful consideration, we feel that it has merit but does not fully meet PLOS Mental Health’s publication criteria as it currently stands. Therefore, we invite you to submit a revised version of the manuscript that addresses the points raised during the review process.

We look forward to receiving your revised manuscript.

Kind regards,

Lambert Zixin Li, Ph.D.

Academic Editor

PLOS Mental Health

Journal Requirements:

1. Please clarify all sources of funding (financial or material support) for your study. List the grants (with grant number) or organizations (with url) that supported your study, including funding received from your institution.

2. State the initials, alongside each funding source, of each author to receive each grant.

3. State what role the funders took in the study. If the funders had no role in your study, please state: “The funders had no role in study design, data collection and analysis, decision to publish, or preparation of the manuscript.”

4. If any authors received a salary from any of your funders, please state which authors and which funders.

2. We have amended your Competing Interest statement to comply with journal style. We kindly ask that you double check the statement and let us know if anything is incorrect.

Additional Editor Comments:

Dear Authors,

Thank you for your submission. Based on the reviewers’ comments and my assessment, the manuscript requires substantial revisions before it can be considered further. Please revise the paper thoroughly and provide a detailed point-by-point response to all comments.

Sincerely,

Lambert Zixin Li, PhD

Reviewers' comments:

Reviewer's Responses to Questions

**Comments to the Author**

1. Does this manuscript meet PLOS Mental Health’s publication criteria?

Reviewer #1: Yes

Reviewer #2: Yes

Reviewer #3: Yes

2. Has the statistical analysis been performed appropriately and rigorously?

Reviewer #1: Yes

Reviewer #2: N/A

Reviewer #3: N/A

3. Have the authors made all data underlying the findings in their manuscript fully available (please refer to the Data Availability Statement at the start of the manuscript PDF file)?

Reviewer #1: Yes

Reviewer #2: Yes

Reviewer #3: No

4. Is the manuscript presented in an intelligible fashion and written in standard English?

Reviewer #1: Yes

Reviewer #2: Yes

Reviewer #3: Yes

Reviewer #1: Thank you for the opportunity to review your manuscript. This is a good and timely paper that will add to the literature on youth mental health. However, the manuscript needs some major revisions. Suggestions and comments are attached below.

General comments:

1. The paper is written in passive voice, which makes it difficult to read.

2. There are no line numberings for the results section after table 1. Needs to work on it.

Abstract:

Line 23 - Provide full text for YMHFA rather than acronyms. Provide full text for PRISMA-ScR if possible.

Line 25 - Sentence is incomplete.

Introduction: The introduction needs a little more work. Add about two paragraphs to deeply engage with works on YMHFA. What you have now is too shallow.

Line 36 - You mention the program is structured around The Action Plan. What is the action plan? We need context.

Line 43 - Statement needs a citation.

Methods:

Line 81 - Use eligibility rather than participants.

Line 97 to 103 - Has this not been discussed already? Content in this subsection seems to have been discussed earlier. Check the subsection on participants. Perhaps combine them?

Line 104 - Just use "Source"

Line 105 to 114 - This is same as above. Combine these subsections and rewrite.

Line 122 to 128 - Same as above comment. It would be best to have a subsection titled "Inclusion and Exclusion" and put everything about inclusion and exclusion in there.

Line 131 to 146 - These subsections can be added to the subsection titled "Search Strategy." Maybe a different title after combining would be helpful.

Results: Needs an introduction.

Line 148 to 154 - These two subsections should be in the Methods section and maybe combined with the appropriate subsections. They do not seem to belong in the Results section.

Facilitators of Implementation - Each point raised in this subsection needs a little more context and explanation.

Barriers to Implementation - Each point raised in this subsection needs a little more context and explanation.

Discussion: The discussion needs a little more description to flesh out your argument. As it stands, it is too shallow. We need to see your argument on what your review says.

Conclusion: I think conclusion should come before "Gaps in literature"

Reviewer #2: Thank you for this research. I found the scoping review of mental health first aid for adolecents timely and informative. The scoping study is well laid out and informative beyond the study itself related to implementation and evaluation of mental health programs.

Reviewer #3: The authors conduct a scoping review of YMHFA programmes for adolescents for research published post 2019. Overall, I think the paper is a good contribution to our understanding on how YMHFA programmes can and are implemented alongside commonly observed outcomes. However, I have a few concerns about the manuscript.

Major comments:

1) YMHFA is the focus of this paper, yet the introduction on what encompasses YMHFA is abrupt and insufficient. While a full narrative review is not needed here, I think much more detail is nevertheless still needed in the introduction for readers who may be unfamiliar with YMHFA and ALGEE. I would recommend including some specific examples of implementation and outcomes here, as well as an overview of the history of these programmes.

2) The authors have included a comparison with a former systematic review in the introduction, but it's not sufficient. The authors need to justify if the goal is simply to synthesize post-2019 evidence, in which case an updated systematic review would be better, but instead the authors conduct scoping review, because this field now has new directions or niche articles that cannot be fully evaluated through systematic review. Accordingly, the authors need to provide more justification of why these dimensions needs a scoping methodology from mapping complexity, heterogeneity, emerging domains, or insufficient evaluative coherence etc. Additionally, the authors should identify new developments since 2019, and explicitly state how the scope of this review diverges from the 2019 systematic review.

3) In the same line, why would the authors conduct a scoping review, instead of a more comprehensive systematic review, if the aim is to “provide a comprehensive understanding of YMHFA’s role in youth mental health support”? This seems to me a question better answered by a systematic review than a scoping review.

4) The authors provide a definition of mental health literacy, implementations. I think some examples and citations to back up these definitions would be helpful. Moreover, as these are the basis for coding the articles, I think much more justification is needed to show the importance of these elements to research on the effectiveness of YMHFA programmes. For example, why focus on effectiveness – which is on intended outcomes, instead of a broader area of outcomes – which may include intended and unintended effects?

5) I think it is not sufficient that the lead author reviewed findings iteratively without any formal inter-rated reliability procedures. Can there be an actual formal procedure, or at least a formal evaluation on the agreement with an external rater, and can there be more information on how the screening team was calibrated, from the pilot screening phase, training, to standardisation procedures, and in doing so, how were disagreements managed?

6) The authors have included a lot of literature which are very heterogeneous, so you need to analyse how heterogeneity affects interpretation, either in the discussion or results sections. For example, how would RCT vs program evaluation affects strength of evidence? How would school vs community-based implementation produce different barriers/facilitators? Are there any different findings from virtual vs in-person?

7) For outcomes, I noticed that much of the improvement in outcomes were focused on adults involved in adolescent mental health (schools, community, parents etc.). While this is great, were there any direct impacts on improvements in adolescent participants’ mental health?

Minor comments:

1) The abstract states that the evidence gathered in this review is from US-based sources, yet in the methods it states that the geographic context was not restricted, but language was also limited to English-based articles. Does that mean that only US-based articles were identified (unlikely since there are studies from Australia and the UK)? Or were there some inconsistencies in between the abstract and methods sections?

2) I think including information on sample size in the summary table would be helpful, and basic descriptives like age and gender. This may be especially important since outcomes may differ between adolescents, that are the explicit focus of the review, and adults who support or interact with target adolescents groups and are still included in the review.

3) I think a differentiation needs to be made between peer-reviewed journal articles and grey literature (policy documents) which may not have been peer-reviewed in the summary table.

**Do you want your identity to be public for this peer review?** For information about this choice, including consent withdrawal, please see our Privacy Policy

Reviewer #1: No

Reviewer #2: No

Reviewer #3: No

---

## [Editor Report · Decision Letter 1]

12 Jan 2026

A scoping review of youth mental health first aid for adolescents in school, community, and healthcare settings.

PMEN-D-25-00530R1

Dear Dr. Alam,

We are pleased to inform you that your manuscript 'A scoping review of youth mental health first aid for adolescents in school, community, and healthcare settings.' has been provisionally accepted for publication in PLOS Mental Health.

Best regards,

Lambert Zixin Li, Ph.D.

Academic Editor

PLOS Mental Health

Dear Authors,

Thank you for your revision. I have carefully assessed your revised manuscript and responses, and I am pleased to accept the manuscript for publication.

Sincerely,

Lambert Zixin Li, PhD